# Physician’s Perspectives on Factors Influencing Antibiotic Resistance: A Qualitative Study in Vietnam

**DOI:** 10.3390/healthcare11010126

**Published:** 2022-12-31

**Authors:** Khanh Nguyen Di, Sun Tee Tay, Sasheela Sri La Sri Ponnampalavanar, Duy Toan Pham, Li Ping Wong

**Affiliations:** 1Department of Academic Affairs—Testing, Dong Nai Technology University, Nguyen Khuyen Street, Trang Dai Ward, Bien Hoa City 810000, Dong Nai, Vietnam; 2Department of Social and Preventive Medicine, Faculty of Medicine, Universiti Malaya, Kuala Lumpur 50603, Malaysia; 3Department of Medical Microbiology, Faculty of Medicine, Universiti Malaya, Kuala Lumpur 50603, Malaysia; 4Department of Medicine, Faculty of Medicine, Universiti Malaya, Kuala Lumpur 50603, Malaysia; 5Department of Chemistry, College of Natural Sciences, Can Tho University, Can Tho 900000, Vietnam

**Keywords:** antibiotic resistance, physician’s perspective, qualitative study, Vietnam

## Abstract

(1) Background: The antibiotic resistance (ABR) rates are escalating to seriously high levels worldwide. This study was conducted to determine physicians’ perspectives on factors influencing ABR in Vietnam. (2) Methods: Focus group discussion (FGD) was conducted through in-depth interviews on ABR perspectives with 5–6 physicians from different geographical locations and hospitals in Vietnam between March and June 2020. The research questions were focused on three main themes of (a) knowledge deficiency on ABR and hospital-acquired infection, (b) antibiotic prescribing practice among clinicians in the healthcare setting, and (c) regulations and hospital policies on antibiotic use. The descriptive analysis was performed using QRS NVivo software. (3) Results: A total of six FGDs were conducted among 34 physicians (18 males, 16 females) aged 26–53 years old from six public and six private hospitals in Vietnam. Most of the participants were attending physicians (85.3%) and had 5–10 years of experience in surgical wards (55.9%). For theme (a), a majority of participants agreed that they had adequate information updates on how ABR develops in their clinical setting; and were well aware of hospital-acquired infections. For theme (b), the participants agreed that WHO guidelines and Vietnam national guidelines were two important reference documents in guiding physicians in antibiotic use. For theme (c), the FGD study revealed awareness of ABR, hospital antibiotic policies, and procedures for administrators on antibiotic use that were updated and complied with. (4) Conclusions: While different levels of control measures against ABR are ongoing in Vietnam, several weaknesses in the current antibiotic prescribing strategies in the hospital and clinical setting management policies have been identified in the healthcare system. The research findings will be helpful for policymakers to have better plans of action against ABR in Vietnam.

## 1. Introduction

With the emergence of multidrug resistance in bacterial strains, the development of antibiotic resistance (ABR) has become a globally recognized human health threat [1,2,3,4]. The overuse of antibiotics has been identified as a major contributory factor leading to the development of antibiotic resistance [5]. The rational use of antibiotics is important to minimize the emergence and spread of resistance [6]. Meanwhile, inappropriate use of antibiotics is known to arise from a complex interplay of several factors, including the prescribers’ knowledge and experiences, diagnostic uncertainty, perceptions of patients in relation to patient-prescriber interaction, and insufficient patient education by physicians [1]. Studies have also indicated the occurrence of unnecessary antibiotic prescribing practice [6,7,8]. Thus, the control of antibiotic usage needs multifaceted interventions involving knowledgeable and engaged healthcare practitioners and the public [6,7,9].

Vietnam, similar to many emerging economies, faces a considerable amount of challenges in attempting to control ABR [7,8]. A high burden of respiratory tract and bloodstream bacterial infections, which require frequent use of antibiotics, has indirectly increased the level of ABR in Vietnam [10]. Lower respiratory tract infections were listed as the 9th leading cause of death in Vietnam recently [9]. In short, ABR has been regarded as one of today’s most challenging problems in medical science, and if the problem persists, it may lead to many clinical, economic, and public health implications. Approximately 90% of Vietnamese pharmacies dispense antibiotics without a prescription [9]. It has been reported that 87% of Vietnamese can easily purchase antibiotics from the pharmacies, hence, it is not surprising that antibiotic resistance is a major public health concern in Vietnam [11]. This may be a factor affecting antibiotic resistance in the community, but within the hospital framework, it has not been thoroughly investigated. Only 27% of the pharmacy staff in Vietnam have correct knowledge about antibiotic use and resistance [12]. This percentage was considered low compared to the results obtained in Jordan (>87%), Norway (57%). Furthermore, nonprescriptive abuse is also a common phenomenon in this developing country. Previous studies in Vietnam reported that prescribing practices were poor (approximately one-third of the patients had an inappropriate indication for prescription) and self-medication was common—often being the most affordable way to access healthcare [7,8,11,13]. Since physicians could significantly affect and influence the patients, their families, and their communities in using antibiotics correctly, it is important to investigate the physician’s perspective on these issues.

To date, the physician’s perspective on the factors influencing ABR in Vietnamese hospitals has not been investigated. The objectives of the study were to determine a physician’s perspective on factors influencing ABR in Vietnam. The findings of this study will help Vietnamese policymakers plan and establish future interventions to improve antibiotic prescriptions in healthcare facilities. Findings will also provide insights to future educational campaigns to improve communication about antibiotic usage between physicians and patients.

## 2. Materials and Methods

### 2.1. Participants

A qualitative approach using focus group discussion (FGD) was conducted between March and June 2020 in six private and six government hospitals in the northern, central, and southern of Vietnam. Physicians that participated in the FGD were Vietnamese, age between 25 and 60 years old, and who had been working in both private and public hospitals in Vietnam. Basically, the urologists, the pediatricians, the ear-nose-throat (ENT) specialists, as well as general practitioners from both outpatient care and inpatient care were invited to participate in order to enhance the diversity of the participants. The physicians varied by year of experience, geographic region (rural, urban), and position of authority in their hospitals. The purposive sampling was solely used to recruit 5–6 physicians for each group based on the inclusive data, totaling five to six groups. The transcripts of each FGD were translated from Vietnamese into English by native speakers who are experts in pharmacy practice areas, and reviewed simultaneously to evaluate whether the thematic saturation point has been achieved or more interviews are to be scheduled. The saturation point was defined as different themes that were repeated many times in an interview and no other themes emerged at all [14]. After six FGDs had been created, the researchers assessed whether data saturation had been reached and the data categories had been created.

The recruitment process was based on the inclusion criteria using snowball sampling, whereby only eligible physicians were invited to join the FGD. The sampling was chosen according to the following criteria: (a) physicians who volunteered to join in the study (volunteers), and (b) physicians of different geographic locations and hospitals (convenience sampling). Participants in each group were healthcare practitioners from both outpatient and inpatient care. Moreover, both male and female physicians and those with low (less than 5 years of experience) or high experience (5–25 years of experience) were invited to participate in each group of the interview to maintain heterogeneity.

### 2.2. Instruments

An interview guide consists of open-ended questions that were used to collect data for this qualitative study. In the semi-structured interviews, the moderators collected answers and ideas from the respondents using the interview guide as a framework. The interview guide was developed based on our preliminary studies and the literature and consisted of three sections, including questions regarding (1) knowledge deficiencies on ABR and hospital-acquired infection, (2) antibiotic prescribing practice among clinicians in the healthcare setting, and (3) regulations and hospital policies on antibiotic use. The interview guide was pilot tested by one of the FGD participants to assess whether it was adequate and relevant for conducting future FGD interviews. Since there were no further revisions or changes needed, the interview guide was used as the final version and for all (six) FGDs for data collection.

### 2.3. Data Collection

Data collection was conducted based on the FGD interview guide described earlier. Six FGD interviews were conducted in the Vietnamese language to get a more in-depth picture of the research matter and to validate the results of the questionnaire, and explore participants’ views in more depth. The objective of the study, as stated in an information sheet, was distributed to all participants in each FGD. The participants signed a consent form before participating in an audio-recorded individual interview. Participants were also required to answer questions regarding their sociodemographic information. Interviewees’ privacy and confidentiality were protected by making their comments anonymous. Honorariums were given to the participants for their time accordingly. Additionally, the original recordings were deleted after the transcriptions were completed.

FGD interviews were conducted either in an authorized office room at Dong Nai Technology University or in a hospital conference room for the convenience of all participants. Discussions were moderated/carried out for approximately one hour, audiotaped, and transcribed from Vietnamese into English. Notes were also taken by researchers and supplemented with audio records to glean details from the discussion. The identities of all participants were kept anonymous during the FGDs and reporting process.

During the discussion, follow-up questions in the interview guide were used by moderators to probe further as well as elucidate/seek clarification from the interviewees. The moderators terminated the discussion when all interviewees agreed that there was no additional information and that the objectives of the interview had been achieved. The saturation point would be reached after six FGD interviews had been completed.

### 2.4. Data Analysis

Transcripts were translated from Vietnamese into English and were peer-reviewed. The transcripts were put into QRS NVivo qualitative software (QRS International Pty Ltd., Doncaster, Victoria, Australia) to identify key themes and patterns for initial analysis. Themes identified after a coding framework (such as summarizing the dominant themes or presenting similarities and differences in related codes to generate meaning) were developed.

## 3. Results

A total of six FGDs were conducted with the voluntary participations of 34 physicians from many hospitals. Table 1 shows the demographic characteristics of the FGD participants. The mean age of the participants was 36.6 ± 1.3 years (range: 26–53 years old). Males and females were equally distributed among the participants, with 52.9% males (*n* = 18) and 47.1% females (*n* = 16). Most participants (55.9%, *n* = 19) had been working as physicians in their respective hospitals for 5–10 years. Most of the participants 85.3% (*n* = 29) were attending physicians, and five were residents, of whom 70.6% (*n* = 24) were serving in the surgical wards. More than half of the participants (67.6%, *n* = 23) were specialists in their hospitals.

Three major themes were identified from the 06 FGDs, e.g., (i) knowledge of ABR and hospital-acquired infection among the physicians; (ii) antibiotic prescribing practice of clinicians in healthcare settings; and (iii) regulations and policies on antibiotic use in hospitals (Table 2).

### 3.1. Theme 1—Physicians’ Knowledge on Antibiotic Resistance and Hospital-Acquired Infection

The physician’s knowledge of ABR and hospital-acquired infection was investigated. Various reasons for inadequate (lack of awareness) knowledge on ABR and hospital-acquired infection that possibly affect the emergence of ABR among the public were discussed. There were six subthemes on the physicians’ knowledge on antibiotic resistance and hospital-acquired infection, e.g.,

(i)information on how antibiotic resistance develops in current clinical settings;(ii)information on how hospital-acquired infection (HAIs) develops in current clinical settings;(iii)common sources of information on antibiotic use in hospitals;(iv)causes of knowledge deficiency among physicians that can contribute to antibiotic resistance;(v)impacts of antibiotic resistance in current clinical settings and how to keep up to date with new developments in clinical field.

#### 3.1.1. Information on How Antibiotic Resistance Develops in Current Clinical Setting

Most of the participants (88%) agreed that they and their colleagues had sufficient information on how ABR develops in their hospitals.

“*The attending physicians in our hospital have sufficient information on how antibiotic resistance develops in the current clinical setting, as evidenced by the fact that in the course of their works, under the management and supervision of the hospital’s board of directors, they provide proper prescription.*”FGD 1, participant 4

“*For the prescribing and dispensing of drugs, especially antibiotics, during prevention and treatment purposes, our medical team has proper prescription recorded, one of them has involved in teaching at some medical universities in and outside the region, and has the opportunity to participate in medical research projects/activities.*”FGD 5, participant 1

“*Apart from diagnosis and treatment, physicians and medical staffs, in general, also participate in research projects and publish articles in scientific journals on medical specialties annually.*”FGD 2, participant 3

Throughout the FGDs, there were no discussions about whether the specialists had provided the wrong antibiotic prescription or given the wrong consultation to patients. There was no statement mentioned that the participants had insufficient information on how ABR develops, except for one comment, as *“This is controversial antibiotic resistance is constantly changing, depending on the hospital and each locality. Up to this point, we have not had any hospital focusing on research on this issue, and only used general materials in the world and in Vietnam at central hospitals, but the problem lies in the fact that, not all documents are suitable for each locality.”* FGD 6, participant 1. This suggests that each local hospital should develop its own guidelines for promoting appropriate antibiotic use. To accomplish this goal, it is also necessary to have the support of national and local health authorities, as well as coordination and cooperation between hospitals. 

#### 3.1.2. Information on How Hospital-Acquired Infection (HAIs) Develops in Current Clinical Setting

All FGDs discussed how hospital-acquired infections (HAIs) develop in their workplace. The participants stated that most of the hospital’s physicians were well aware of this issue. They had proper prevention for themselves and their patients; accurate diagnosis with the symptoms of HAIs, and a proper antibiotic prescription. The design and arrangement of hospital supplies were well managed by hospitals to avoid the growth and spread of bacteria. Airflow in the operating room was regulated using two-way circulations (in and out) for separate areas: one for operating room staff to change clothes before entering the operating room area, and the operating room area itself. Programs for monitoring hospital infections were seriously implemented. Disinfection and sterilization of the hospital were carried out very strictly. In addition, basic hygiene for medical staff is also strictly enforced.

“*There were physician’s awareness of nosocomial infections and compliance, e.g., regular hand hygiene, neat clothing and hair during work and before entering operating room. The design of operating rooms, airflow in and out, and cleaning of hospital utensils were also well done.*”FGD 1, participant 3

“*Currently, our hospital has an infection control department with the role of monitoring and disseminating information periodically through weekly and monthly briefings for all hospital staff. When there is an infection in any department, they will notify other departments.*”FGD 4, participant 2

There was no statement that the physician in their clinical setting had insufficient information on how HAIs develop.

#### 3.1.3. Common Sources of Information about the Use of Antibiotics in Hospitals

Most of the FGD participants mentioned that the practice guidelines issued by the Vietnam Ministry of Health (MOH), in 2015 are a common source of information on antibiotic use in their hospitals. Some of them mentioned that they could also acquire relevant information from various internet sources, and by attending conferences, workshops, seminars, and pharmaceutical advertisements (advertisements in books, newspapers, magazines, on radio and television, in electronic newspapers, etc.), for continuing medical education. A few of the participants mentioned that they could find updates in special journals.

“*Our hospital refers to the “Instructions on Antibiotic Use” published by the Ministry of Health (issued with Decision no. 708/QD-BYT dated 2 March 2015) for antibiotic use. The process of prescribing antibiotics is mainly based on this guideline, however in some special cases, we refer to other documents including guidelines from foreign medical companies such as Stanford, Medscape, combined with documents of Central hospitals and after consultation among different specialists.*”FGD 4, participant 1

“*Depending on each case, before prescribing antibiotics to patients, I follow the guidelines of the Ministry of Health and the guidelines of the central hospital. In addition, for some special cases, there must be consultation with the professional directorate, pharmacy department, internal medicine department, and emergency department.*”FGD 4, participant 2

“*The source of information about new antibiotics is updated based on the recommendations of pharmaceutical companies, but it is also somewhat limited. For myself, as well as my colleagues in the faculty, the source of information is obtained by attending seminars. In addition, the pharmacy department of our hospital is the unit to access and disseminate updated information on antibiotics to other departments.*”FGD 6, participant 5

When asked to suggest solutions to help specialists and clinical staff acquire up-to-date information on antibiotic usage and resistance, several ideas were raised and discussed. They expressed their desire to establish research groups on the state of antibiotic use in inpatient and outpatient treatment at the national level. They are aware that the problem of ABR is very complicated, as 

“*Antibiotic resistance is complex and continuous; we need to have research on each location and detailed statistics. The establishment of national-level in-depth research groups on the topic of antibiotic resistance is necessary. Because when participating in such research, we will have the opportunity to learn and exchange information with each other, thereby serving the diagnosis and treatment of diseases as well as having effective antibiotic prescription. Besides prescribing based on antibiogram results, empiric antibiotic prescribing is also effective in treatment when the physician has good experience.” FGD 4, participant 6; and “I propose the establishment of national-level research projects on antibiotic use and antibiotic resistance. If we can implement such topics, our hospital’s medical team will have the opportunity to learn and update information in a timelier manner.*”FGD 5, participant 2

#### 3.1.4. Causes of Knowledge Deficiency among Physicians That Can Contribute to Antibiotic Resistance

The FGD participants expressed their opinions on the causes of how limited knowledge might hamper the prevention of antibiotic resistance. The common causes include a lack of self-updates, and a lack of regular and advanced training from hospitals. There were no comments from the physicians on the comprehensive infection control programs available, and/or antibiotic stewardship strategies available. Only a few of the medical staff complained about the lack of opportunities to attend conferences and continued their education until they graduated as medical doctors. In addition, some also expressed their desire to receive online teaching activities and provision of written national guidelines. 

“*The lack of knowledge among medical professionals in the use of antibiotics causes antibiotic resistance in the community. The first reason is that they do not update information themselves, and the second one is that the use of antibiotics in treatment is not completely according to patient’s condition but according to other benefits (e.g., commission for prescribing). Usually, a majority of doctors work under the direction of the hospital management board, following the rules and regulations.*”FGD 1, participant 1

“*At our hospital, the connection and transfer of experience between generations of doctors is very good, they share experiences for the younger generation, and learn from each other. Therefore, this should not be the cause of the lack of knowledge about antibiotic use among physicians. I think the main reason is the lack of regular training, advanced training, and more importantly, the sense of self-updating of each doctor so that they don’t become obsolete.*”FGD 4, participant 6

“*In order to prescribe antibiotics to patients according to the results of microbiological tests or antibiograms, we need a certain time of 5–7 days, thus, the answer of whether or not doctors who prescribed antibiotics according to experience is “yes”, in fact.*”FGD2, participant 2

“*Not being able to participate in lectures, and research projects is not the cause of the lack of knowledge about antibiotics and antibiotic resistance.*”FGD 6, participant 1

There were suggestions to make physicians more updated and knowledgeable to antibiotic use and resistance to prevent ABR, For instance, by joining a national project on antibiotic use and antibiotic resistance, writing and publishing journals, and/or presenting their research findings in seminars or conferences.

“*Some physicians excessively care about their family’s finances. In Vietnam, the financial resources for doctors are quite limited, so they look for a high-paying job, and of course high salary is equivalent to a lot of work, thus, they do not have adequate time to update their knowledge. Additionally, if a physician work at a low-wage state hospital, he/she will find a part-time job, so he/she does not have time to update their knowledge, hence, in order for them to have enough time to update their knowledge, salary policy should be changed by government.*”FGD 3, participant 3

“*I think in order for the medical team to have up-to-date knowledge on antibiotic use, the hospital’s director board should create more opportunities for them to participate in specialized seminars and conferences as experts or speakers. At the same time, every year the hospital should encourage them to participate in research and publish articles in scientific journals.*”FGD 6, participant 2

#### 3.1.5. Effects of Antibiotic Resistance in Current Clinical Settings

Most of the FGD participants did not discuss much about the impact of ABR in their current clinical settings. In some FGDs, the participants expressed their concerns about the impacts, including the increase in morbidity and mortality, more difficulty in treating the basic infectious condition, the high cost of treatment, and the need for patients to spend more time in the hospital to cure the disease. 

“*For patients whose infections are resistant to antibiotics, it becomes more difficult to use antibiotics to treat the infections. In such situation, we have to be careful in performing microbiological and antibiotic susceptibility tests, in order to choose the most optimal treatment regimen for the patient. It usually takes longer time to treat these resistant infections. The cost of their treatment is also higher than usual compared to other patients who respond well to the initial course of antibiotics.*”FGD 1, participant 2

“*This puts a heavy burden on both the patient and hospital, when the length of treatment is prolonged. For patients infected with multi-drug resistant bacteria, our hospital’s doctors will put them in an isolation room to limit cross-infection with other patients with weak resistance.*”FGD 2, participant 1

#### 3.1.6. How to Keep Up-To-Date with New Developments in the Clinical Field

In Vietnam, each clinical setting has identified several approaches to help physicians and medical staff keep abreast with new developments in the clinical field. It is revealed that some hospitals encourage their staff to attend internal and external conferences, seminars, and workshops, while some others provide resources and reading materials, such as specialized documents, books, and journals.

“*In order for medical staff to stay up-to-date with new developments in their field, I think they should be given more opportunities to participate in conferences and seminars within and outside the region, but in reality, in my hospital such opportunity is rare.*”FGD 5, participant 1

“*Currently, specialized documents are updated but not regularly. I think, update frequency should be weekly or monthly. However, it is difficult to implement in the current context of both the provincial and central hospitals, the currently every year update is good enough.*”FGD 6, participant 3

### 3.2. Theme 2—Antibiotic Prescribing Practice among Clinicians in Healthcare Setting

Realizing the importance of medicine prescribing, especially antibiotic prescribing practice among clinicians in a healthcare setting, national guidelines are made available to Vietnamese hospitals to prevent/reduce indiscriminate prescriptions in order to develop outpatient treatment protocols, manage inpatient treatment regimens, and limit drug abuse. Factors influencing antibiotic prescribing practices in outpatient and inpatient care were investigated in this study.

#### 3.2.1. Factors Affecting the Prescription of Antibiotics in Outpatient Care

Antibiotic prescribing practices in outpatient care can lead to antibiotic resistance in clinical settings. This is one of the most discussed parts of all FGDs. The participants expressed their thoughts about various factors that would influence antibiotic prescribing practices in outpatient care. According to the participants, the process of making decisions on antibiotic prescribing should adhere strictly to hospital guidelines, rules, and regulations. In some cases, empirical treatment was given to patients before the results of antibiotic susceptibility tests were received. Physicians’ experience, patients’ health history, financial status, and patients’ expectations are other factors affecting the prescription of antibiotics in outpatient care. Patient compliance was also one factor that influenced their antibiotic prescribing practice for outpatients. Among the factors that have been identified, physicians’ experiences were identified by almost half of the physicians as the most relevant.

“*Discussing the factors that affect antibiotic prescribing in outpatient care, the first thing I want to say is that based on experience in prescribing antibiotics, a doctor can recognize whether an antibiotic is consistent with the causative agent in the patient or not. Next is the patient’s financial ability, in fact, each antibiotic has a different price, so depending on the patient’s economic condition, the doctor will have a suitable choice.*”FGD 1, participant 3

“*Depending on the patient, if they don’t have money, our doctors will advise and prescribe low-cost antibiotics. That is, when advising on antibiotics, we advise on the effects of the drugs along with their prices, so if the patient is not economically eligible, the doctor will prescribe a cheaper antibiotic.*”FGD 2, participant 4

“*In determining the treatment regimen, for example, when a patient with pneumonia was admitted to the hospital, the doctor will initially prescribe a beta-lactam antibiotic, or a second or third-generation of cephalosporin (e.g., cefixime, ceftibuten, cefpodoxime), plus a common macrolide or an antibiotic used to treat atypical bacteria, and gram-positive bacteria. Currently, as bacteria are completely resistant to the macrolide group of antibiotics, the doctor will not prescribe such regimen anymore, instead; they will prescribe a third-generation cephalosporin in combination with another antibiotic to improve treatment outcome because the rate of bacterial resistance has already become too high.*”FGD 3, participant 5

“*Following the words of a colleague, I also want to share another influencing factor, related to the drug stock. When the hospital’s available drug stock does not meet the requirements, it is difficult to choose the desired antibiotic for patients. Honestly, when a certain pharmaceutical brand enters the hospital to present the drug and introduce it, they need to go through many “backdoors”. The quality of drugs between brands is of course not the same. In terms of pharmacology, we do not discuss, the dosage levels are similar. However, the time of release and the amount entering the blood of patients when using such drugs are different, effectiveness in treatment hence also different. We ourselves, when prescribing antibiotics, might be governed by the drug list available in the pharmacy store of the hospital.*”FGD 6, participant 3

“*The factors listed below are all factors that influence antibiotic prescribing practice: patients’ health history, current clinical conditions, patient demand for antibiotics, physicians’ experience on antibiotic prescribing, cost of treatment, patients’ compliance, physicians’ experience in treating the infectious diseases, antibiogram availability for the physicians to assess antibiotic susceptibility and select empirical antibiotic treatment.*”FGD 2, participant 1

#### 3.2.2. Factors Affecting the Prescription of Antibiotics in Inpatient Care

Similar to outpatient antibiotic prescribing routine, antibiotic prescriptions for in-hospital patients were also influenced by factors, such as antibiogram availability, experience of physicians with infectious diseases, patients’ health history, and current clinical conditions. Physicians agreed on how antibiotic prescribing practices for inpatient care might lead to the emergence of ABR in their current clinical settings. Hence, some physicians expressed their foreseen weaknesses or limitations in their current antibiotic prescribing strategies in their hospitals and gave suggestions for further improvement. 

“*When prescribing antibiotics to outpatients, I have to consider a lot about the patient’s history, their current medical condition, whether the antibiotic is appropriate or not. For example, if the patient’s liver and kidneys have problems, it is not suitable to use that antibiotic, I have to change to another suitable one.*”FGD 1, participant 1

“*The economic status of the patients was one of the factors influencing my choice of antibiotics for their treatment.*”FGD 1, participant 5

“*When I want to prescribe certain antibiotic to a, but the hospital pharmacy department doesn’t have that antibiotic, I have to change it to another one. Therefore, the list of antibiotics available in the hospital pharmacy is also one of the influencing factors.*”FGD 4, participant 3

“*The physician’s prescription has impacts on the antibiotic resistance among people in the community, because like outside the community, there is also abuse (indiscriminate use) of antibiotics inside the hospital. For example, even if the status of the patient is not up to the point of needing antibiotics, he/she is still taking antibiotics because of some influencing factors. That is real and happening in my workplace. In addition, in some cases patients with recurrent infections, the physicians have to use many different antibiotics due to changing treatment regimens. This gives rise to multi-antibiotic resistance.*”FGD 5, participant 1

“*Doctors prescribe antibiotics by experience, of course, for inpatients, we all do a test (sputum culture) before prescribing antibiotics, especially injectable antibiotics. As for oral antibiotics, it is not necessary to culture sputum to make an antibiogram.*”FGD 3, participant 1

“*The use of antibiotics in the community is very indiscriminate, an ordinary citizen can go to any pharmacy to buy antibiotics at any time. When they use antibiotics ineffectively, they will go to the hospital, the application of escalation/de-escalation of antibiotic will no longer be appropriate because it takes time, and the bacteria might have become resistant. In private hospitals, doctors use moderate or high levels of antibiotics, e.g., strong antibiotics, higher than what the patient used before admission, of course but not as high as in the big central hospitals.*” FGD 3, participant 4

### 3.3. Theme 3—Regulations and Hospital Policies Regarding the Use of Antibiotics

One of the causes leading to ABR was the widespread, inappropriate use of antibiotics without a medical prescription, making them less effective or ineffective. According to the survey results on the sale of antibiotics at drug retailers in rural and urban areas in the northern provinces, the awareness of antibiotics and ABR among drug sellers and people is particularly low, especially in rural areas. Most antibiotics are sold without a prescription, 88% in urban areas and 91% in rural ones [11]. Two subthemes include hospitals’ compliance with regulations of the use of antibiotics and their policy on the use of antibiotics.

#### 3.3.1. Regulation of the Use of Antibiotics by Hospitals

Strengthening the control of antibiotic prescription and selling prescribed antibiotics are two strategies recommended by the WHO and Vietnam MOH for reducing ABR. Decision 4041/QD-BYT of the MOH on “Approval of the project to strengthen the control of antibiotic prescription and the sale of prescribed antibiotics for the period 2017–2020”, in order to raise public awareness about the proper use of antibiotics, and enhance the responsibility of professional clinicians, especially prescribers and retailers, in implementing the law, thereby contributing to reducing ABR, drug abuse, and inappropriate drug use. 

Most of the participants thought that the regulations were strict enough for physicians to comply with in order to control ABR. Some expressed their opinions that the surveillance program, the transparency and the feedback on antibiotic consumption, costs, and trends in the hospital setting were essential in preventing ABR. The physicians agreed that regulations of the use of antibiotics contribute to ABR control in their clinical settings. In addition, based on their own prescribing experiences, they foresee gaps and weaknesses that need to be improved in the current regulations regarding the use of antibiotics.

“*Our hospital always adheres to the regulations on the use of antibiotics. This is done strictly. In the case that physicians use antibiotics for prevention and treatment purposes for patients improperly, the incident will be recorded by the management boards and put in the monthly performance evaluation, not just a mere reminder.*”FGD 1, participant 3

“*In this regard, I certify that our hospital is complying with the regulations of the Ministry of Health. I also comply well. However, not all physicians or medical staffs performed well, for many different reasons, maybe due to their knowledge, or personal benefits.*”FGD 4, participant 2

“*In general, our hospital is well managed, the medical staff always strictly follow the regulations of the MOH. However, there are also flaws and inadequacies. Firstly, on the pharmacy side, they mostly know about pharmacology, while we work on the clinical side and when we do something wrong, they don’t even know. The pharmacy department themselves are not correct in some cases of dispensing drugs to patients due to the limitation of the drug store they manage.*”FGD 5, participant 5

#### 3.3.2. Policy on the Use of Antibiotics by Hospitals

All the participants in the FGDs mentioned that there were policies and procedures for administrators on the use of antibiotics that their hospital was following. Information management policies on the use of antibiotics were strictly implemented and controlled. Most of the participants were aware of how the policies on the use of antibiotics can help in reducing ABR in their clinical setting, under the effective-controlled antibiotic stewardship programs. Nevertheless, participants from the suburban area expressed their opinion on the fact that physicians in their current clinical setting did not comply with existing policies, some said that the management policies were poor in their hospital. From that, there were suggestions from the physicians for the policymakers or governing bodies regarding the use of antibiotics to prevent ABR. 

“*About the policy of using antibiotics, my hospital is very clear. If the doctor does not follow those policies, firstly they will be reminded, then if they still repeat the mistakes, their names will be included in the monthly emulation assessment, or they need to meet the management board directly to resolve the circumstances depending on the level of illegal actions repeatedly.*”FGD 1, participant 2

“*In addition, our hospitals also have software support to report illegal actions and statistics of the use of antibiotics*”FGD 1, participant 3

The suggestion from participants in FGDs were recorded as follow: 

“*Regarding the penalty in the indiscriminate use of antibiotics, it would be great if the government had stricter control policies, especially in rural areas. Because most pharmacies in urban areas sell prescription drugs, not indiscriminately. However, in rural areas, this is not always strictly followed. If we want to apply penalty or sanctions, we implement in the urban area first, then in the suburbs and rural areas. It is necessary to strictly prohibit private clinics and pharmacies to sell antibiotics without a medical prescription, they will be fined, banned from doing business, depending on the government.*”FGD 3, participant 4

“*The most practicality to improve antibiotic prescription and indiscriminate use of antibiotics, in my opinion is that we have to manage the supply sources of the antibiotics, because it is very complicated and rampant currently. Besides, the cost of antibiotics is also a problem worthy of interest, and need more analysis. We look forward to hospitals to ensure that antibiotics with reasonable prices can reach patients as compared to the original one.*”FGD 4, participant 4

“*The procedure of doctors and staff in the policy of using antibiotics in the hospital is not quite reasonable, but more importantly we should do it to train them, helping them update information. Because we have to recognize that, most doctors have strong personality, sanctions for them as an extra catalyst for non-regulated antibiotics during prescriptions.*”FGD 6, participant 2

“*I think hospital side should have updated new documents to provide the medical team timely updates. Especially, for the case of patients with severe infections, we need consultation and optimal treatment protocols best updates from the central or international hospitals.*”FGD 5, participant 1

## 4. Discussion

A qualitative study using the FGD approach has been conducted to assess physicians’ perspectives on factors influencing their antibiotic prescribing practice as well as ABR in both outpatient and inpatient healthcare settings in Berlin, Germany [15]. The results showed differential interests among the physicians about topics, such as antibiotic prescribing and ABR. The outpatient care physicians were interested in talking about patient demand and noncompliance (which have been identified by a previous study [16] as well as their antibiotic prescribing practice [17], while the inpatient care ones cared and talked much about multi-resistant pathogens, limited consult time, local resistance data availability [18], and hygiene challenges. Some other qualitative studies have been done to investigate the physicians’ views on this topic [7,13,19,20,21,22]. The results from their study found factors influencing the antibiotic prescribing decision, such as the clinical situation, advance care plans, utilization of diagnostic resources, physicians’ perceived risks, influence of others, and influence of the environment. In this qualitative study, the FGD approach was utilized to gain an in-depth understanding of the physician’s perspective on various factors influencing the emergence of ABR in the community. Three aspects were deeply discussed by participants in all FGDs, e.g., lack of awareness on ABR and hospital-acquired infection; antibiotic prescribing practice among clinicians in healthcare setting; and regulations and hospital policies on the use of antibiotics. 

### 4.1. Physician’s Knowledge on ABR and Hospital-Acquired Infection

Most of the participants in all FGDs agreed that they and their colleagues had sufficient information on how ABR developed in their hospitals. Some evidence supports their answers, including correct antibiotic prescription to patients and patients’ education, attending teaching sections and medical conferences as an attendee and speaker, participating in medical research projects and works, and publishing articles in scientific journals on medical specialties. In other words, there was no mention that the participants had insufficient information on how ABR develops. Only one comment from FGD 6, participant 1 that the physicians in sub-urban and rural areas might had insufficient knowledge on ABR. 

In fact, limited knowledge about the use of antibiotics and updates on antibiotic guidelines have been reported to hamper prevention and control of ABR in the community [8,11,13]. The physicians in this study were aware of this issue (this should be in the introduction/give appropriate citation). The main reasons for the lack of information update as mentioned in FGDs were due to lack of planning, limited opportunities for updates and trainings/lack of resources. The participants have the opinions that advanced trainings regarding their current clinical settings, and opportunities to attend in specialized seminars and conferences as experts or speakers are important. In particular, one of the comments about the lack of up-to-date knowledge of physicians and health professionals about the indiscriminate use of antibiotics in the community and the alarming situation of ABR (rephrase) were due to the fact that financial constraint (low-income status) of physicians. In other words, they have to work part time (such as work in private healthcare clinic) to ensure a stable income to support their family, so they do not have time to improve and update their knowledge on a regular basis [8], and it is a common problem in developing countries [9,23,24,25]. 

According to the opinions of the FGD participants, the impact of ABR to the current clinical settings is long term and un-surmounted, as it causes an increase in morbidity and mortality, more difficulty in treatment, increased cost and duration of hospital stays. Microbiological investigation and antibiotic susceptibility tests are important to determine the most optimal regimen for the patients, which indirectly minimizing the impact of ABR. In addition, following the antibiotic guidelines and policy, patients infected with multidrug resistant organisms should be placed in an isolation room to limit cross-infection with other patients. The findings obtained in this study suggests high awareness of the physicians on the impact of ABR, which has also been recognized as a public health burden worldwide. 

The participants in this study were also well aware of hospital-acquired infection and its consequences. This is consistent with the results from the study of [23]. They were of the opinions that the physicians and medical staffs in their hospitals had proper prevention and control measures against HAIs, for instance, in prescribing proper antibiotic to patients; implementation of antibiotic stewardship programs; disinfection and sterilization etc. This is understandable, as most hospitals and medical facilities in Vietnam, including central hospitals and private hospitals, are required to comply with government regulations and policies. 

The participants have also identified several common sources of information about the use of antibiotics use in the hospital, which include practice guidelines issued by Vietnam Ministry of Health (MOH), 2015; internet source, medical education, conferences, workshops, seminars, and pharmaceutical advertisements (books, newspapers, magazines, on radio and television, in electronic newspapers, etc.). A few participants mentioned that they could obtain updates on antibiotic use and ABR by referring to certain medical journals. However, in general, they felt that the above sources of information did not meet the need for regular updates on the use of antibiotics and ongoing ABR in Vietnam. There were suggestions made during the discussions to diversify information updates. The participants wish to establish national-level research projects on the use of antibiotics and ABR, so that they would have more chances to learn and exchange information with each other in the diagnosis and treatment of diseases. 

Lack of knowledge and inappropriate extrapolation of knowledge about medicines were common causes of prescribing error. It is therefore important that prescribers have immediate access to quality up-to-date drug information to ensure safe and effective prescribing. The FGD also discussed methods of keeping physicians up-to-date in medical field. Some approaches mentioned including encouraging staffs to attend internal and external conferences, seminar and workshop; and providing resources such as documents, books, and medical journals. Weekly/monthly updates on the use of antibiotics and ABR are essential. These resources are also reported to be important and are used by doctors for prescribing in Ireland [26]. 

### 4.2. Antibiotic Prescribing Practices in Healthcare Settings in Vietnam

The FGD participants revealed the implementation of proper antibiotic use based on guidelines from WHO and Vietnam’s national MOH for development of outpatient treatment protocols, inpatient treatment, and limit drug abuse. In fact, all participants agreed/realized that good antibiotics prescribing practice is an important factor for prevention and control of ABR. 

In different countries, people hold different ideas about health, causes of diseases, coping strategies, and treatment modalities [9]. These ideas shape both the expectations and the behavior of professionals in a country’s hospitals [12]. In hierarchical societies, such as Vietnam, people tend to use more antibiotics in prescription practice, because they expected that a combination of antibiotics would enhance the ability to kill bacteria and improve the outcome of the treatment rapidly [11]. A narrative review showed that antibiotic prescribing was influenced by factors such as the patient (e.g., a high-risk or vulnerable patient history), prescribing physician (e.g., fear of failure, diagnostic uncertainty, or inadequate training), and the environment (e.g., regulation of prescribing and dispensing and lack of resources for etiological diagnosis) [24]. In this study, factors influencing antibiotic prescribing in outpatient care are one of the most discussed parts in all FGDs. Several factors, which could lead to proper and/or improper antibiotic prescribing practice, were discussed, including physician experience on antibiotic prescribing, antibiogram test, patients’ health history and medical conditions, financial issue, specific demand, and patients’ compliance. In summary, understanding of the factors that influence the outpatient antibiotic prescribing practice is important for formulation of strategies to prevent and control ABR in Vietnam. 

Another aspect, antibiotic prescribing practice for inpatient, was also deeply discussed in most FGDs. As compared to outpatient care, prescribing processes for inpatient ones have been more strictly controlled by hospital boards. Factors such as antibiogram availability, experience of the physicians with infectious diseases, patient health history and their current status are mentioned in the participant’s discussion. The physicians also expressed weakness or limitation of their current antibiotic prescribing strategies in the hospital and suggested for further improvement. Although the application of escalation/de-escalation of antibiotics were implemented by the physicians for in patients, however, it may not timely for infections due to multidrug-resistant bacteria. Although, de-escalation strategy is recommended to prevent the emergence of resistant bacteria [3,27] there is evidence that the use of an antimicrobial de-escalation strategy limits its feasibility in the presence of multi-resistant pathogens. 

### 4.3. Physician’s Perspective on Regulations and Hospital Policies on the Use of Antibiotics

Previous studies have shown that some of the leading causes of ABR include the lack of official government policy on the rational use of antibiotics in both public and private hospitals [2,3,13,28,29,30,31].

Physician prescribing behavior is associated with antibiotic misuse. Therefore, antibiotic prescribing behavior should be changed and updated according to the latest guidelines to reduce ABR, which is an issue of particular concern in many countries around the world [24,32]. Regarding regulations on the use of antibiotics in Vietnam, the Ministry of Health has issued national guidelines, together with WHO, to strengthen control of antibiotic prescription and selling OTC medicines. These guidelines help to raise public awareness about the proper use of antibiotics, and enhance the responsibility of healthcare professionals, especially prescribers and retailers, in implementing the law, have direct or indirectly contributed to the efforts in reducing ABR, and inappropriate antibiotic use. 

Most of the participants thought that the regulations were strict enough for physicians to be compliant in order to control ABR. These findings are consistent with the study of [23]. Some expressed their opinions that the surveillance program, the transparency, and the feedback on antibiotic consumption, costs, and trends in the hospital setting were essential in preventing ABR. Most of the physicians expressed that their hospitals are currently strictly following and adhering to the regulations from MOH. The department of pharmacy in their hospitals manages the antibiotic list and controls the antibiotics dispensed to patients. In addition, some hospitals use software to report illegal actions and statistics on the use of antibiotics and resistance rate, drug interactions. They also foresaw gaps and weaknesses that needed to be improved in the current regulations regarding the use of antibiotics. They suggested the MOH should have stricter control policies, especially in rural areas, which usually sell drugs without a doctor’s prescription [32]. It is necessary to strictly prohibit private clinics and pharmacies from selling unprescribed antibiotics. Under the enforcement of MOH, such activities will cause them to be fined and/or banned from doing business. 

It is necessary to strictly manage the supply sources of the antibiotics, because it is very complicated and rampant currently. The cost of antibiotics is also a problem worthy of interest and needs more analysis by legislators or policymakers. Hospitals should ensure antibiotics with reasonable prices can reach patients. More importantly, the management boards should train physicians and medical staffs to update information in the clinical settings as reported in this and previous studies [7,13,24,33]. 

### 4.4. Limitations of Qualitative Study

There are some limitations to this study. The convenience sampling may be biased towards including groups of young physicians who were selected for FGDs. In the interpretation of the results, it needs to be carefully considered before drawing conclusions from the sample selected in the qualitative research phase, given the qualitative nature of this study. Participants were based on southern areas of Vietnam, which means that physicians and medical staff from other geographical locations (such as middle of Vietnam and northern area) might be under-presented. The low presentation of high-experience physicians must be considered. Socially desirable responses of the participants obtained may also be exist due to self-report-answer in the FGDs. Although measures were taken (random coding by peer) to reduce bias, the transcript-coding step may also be another limitation of study. The subjective analysis in the process of interpreting the collected data can be one of the limitations, although neutrality was attempted. Insights from this study can be used as a basic pattern for FGD research design to access the physician’s perspective on the use of antibiotics and resistance among the public. It is highly recommended that future studies should target more on physician’s recommendations and suggestions on the government’s regulations and policies to improve the ABR situation among the public in Vietnam.

## 5. Conclusions

This study highlights a physician’s perspective on antibiotic usage and the emergence of antibiotic resistance in Vietnam using the FGD approach. While different levels of control measures against ABR are ongoing in Vietnam, several weaknesses in the current antibiotic prescribing strategies in their hospital and clinical setting management policies were identified in the healthcare system. Based on the research outcomes, practitioners, policymakers, and the community should work together to find effective solutions and measures to abrogate ABR for a better healthcare system and environment. In addition, the findings should also provide insights into future education-based interventions to improve rational prescribing and use of antibiotics.

## Figures and Tables

**Table 1 healthcare-11-00126-t001:** Demographic characteristics of participants in FGDs (*n* = 34).

Socio-Demographic Variables	*n* (%)
**Gender**	
Male	18 (52.9)
Female	16 (47.1)
**Number of years in practice**	
1–5 year or less	6 (17.6)
5–10 years	19 (55.9)
10–15 years	8 (23.5)
15–20 years	1 (2.9)
>20 years	0 (0)
**Position in hospital**	
Resident	5 (14.7)
Attending physician	29 (85.3)
**Location of hospital/clinic**	
Urban	28 (82.4)
Suburban	6 (17.6)
Rural	0 (0)
**Department**	
Medical ward	5 (14.7)
Surgical ward	24 (70.6)
Gynecology and obstetrics ward	1 (2.9)
Pediatric ward	2 (5.9)
Others	2 (5.9)
**Profession**	
Head of department	4 (11.8)
Specialist physicians	23 (67.6)
Non-specialist physicians	5 (14.7)
Others and Unspecified	2 (5.9)

**Table 2 healthcare-11-00126-t002:** Themes and subthemes of the ABR investigations in the study FGDs.

Theme	Subtheme
Physicians’ knowledge on antibiotic resistance and hospital-acquired infection	- Information on how antibiotic resistance develops in current clinical settings. - Information on how hospital-acquired infection (HAIs) develops in current clinical settings. - Common sources of information about antibiotic use in hospitals. - Causes of knowledge deficiency among physicians that can contribute to antibiotic resistance. - Impacts of antibiotic resistance in current clinical settings. - How to keep up to date with new developments in clinical settings.
Antibiotic prescribing practice among clinicians in healthcare setting	- Factors that influence antibiotic prescribing in outpatient care. - Factors that influence antibiotic prescribing in inpatient care.
Regulations and hospital policies on antibiotic use	- Regulation on antibiotic use that hospitals are following. - Policy on antibiotic use that hospitals are following.

## Data Availability

Not applicable.

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
