# Peer review of "Physician’s Perspectives on Factors Influencing Antibiotic Resistance: A Qualitative Study in Vietnam"

_healthcare, 2022, doi:10.3390/healthcare11010126_

Round 1

Reviewer 1 Report

Thank you for this interesting study- please see below my comments to improve the study:

-          The abstract needs some improvement- please list identified themes (only 2 themes were identified? Or more?); the abstract’s conclusion does not tie well with results presented in the abstract, e.g. regulations is not mentioned in the results of the abstract but it is mentioned in the conclusion.

-          Methods: For this statement; “The transcripts of each FGD were translated from Vietnamese into English and reviewed simultaneously…” how was the translated transcripts validated for ‘accurate translation’?

-          Methods: How was the interview guide devised/ developed?

-          Methods: for this statement; “They were paid for their participation and time accordingly…” , it might be better if you use words like ‘honorarium’, etc.

-          Methods: it would benefit from referencing certain studies/ systematic reviews (if any) to show that saturation point can be reached within 6 FGs.

-          The results section would benefit from summarising the themes and subthemes with important supporting quotes into a Table. In addition, a figure that is linking themes and subthemes and how they are linked with the study objective would help better visualisation of the results.

Author Response

December 18th, 2022

To: Healthcare

Dear Editor and Editorial staffs,

We would like to send the revision letter regarding to the manuscript healthcare-2112814 entitled “Physician’s perspectives on factors influencing antibiotic resistance: a qualitative study in Vietnam”, submitted to Healthcare,by Khanh Nguyen Di et al.

We sincerely appreciate the hard work of the journal and the reviewers, which definitely make our manuscript become better. Please find the responses to reviewers’ comments in the next pages of this letter. All changes were made under “track change” mode in Word processing program.

Thank you very much for your consideration. In case more information is required, please feel free to contact us.

Best regards,

Khanh Nguyen Di

Responses to the reviewers

REVIEWER # 1

- The abstract needs some improvement- please list identified themes (only 2 themes were identified? Or more?); the abstract’s conclusion does not tie well with results presented in the abstract, e.g. regulations is not mentioned in the results of the abstract but it is mentioned in the conclusion.

Thank you. We have revised the abstract accordingly, following your suggestions. The themes were specifically clarified and the results/conclusion have been adjusted.

- Methods: For this statement; “The transcripts of each FGD were translated from Vietnamese into English and reviewed simultaneously…” how was the translated transcripts validated for ‘accurate translation’?

Thanks for your question. The transcripts were translated by native speakers who are also experts in the pharmacy practice field. Moreover, each transcript was translated and reviewed simultaneously by at least 02 different experts. Thus, the accurate translation should be achieved. We have added these information in the manuscript.

- Methods: How was the interview guide devised/ developed?

Thank you. The interview guide was developed based on our preliminary studies and the litera-ture. We also conducted pilot test to validate the guide. We have added more information in this section.

- Methods: for this statement; “They were paid for their participation and time accordingly…” , it might be better if you use words like ‘honorarium’, etc.

Thanks so much for your suggestion. We have adjusted the word accordingly.

- Methods: it would benefit from referencing certain studies/ systematic reviews (if any) to show that saturation point can be reached within 6 FGs.

Thank you. We have added a reference related to this issue accordingly.

- The results section would benefit from summarising the themes and subthemes with important supporting quotes into a Table. In addition, a figure that is linking themes and subthemes and how they are linked with the study objective would help better visualisation of the results.

Thanks for your suggestion. We have added a Table, which describe all relevant themes/subthemes of the study.

Reviewer 2 Report

This paper contains important findings in terms of investigating medical professionals' perception of antimicrobial resistance, which is a global problem, and linking it to education. In this study, the number of participants is small scale, and its evaluation is limited, I believe that there is no problem in present form, and this paper could be accepted.

Author Response

December 18th, 2022

To: Healthcare

Dear Editor and Editorial staffs,

We would like to send the revision letter regarding to the manuscript healthcare-2112814 entitled “Physician’s perspectives on factors influencing antibiotic resistance: a qualitative study in Vietnam”, submitted to Healthcare,by Khanh Nguyen Di et al.

We sincerely appreciate the hard work of the journal and the reviewers, which definitely make our manuscript become better. Please find the responses to reviewers’ comments in the next pages of this letter. All changes were made under “track change” mode in Word processing program.

Thank you very much for your consideration. In case more information is required, please feel free to contact us.

Best regards,

Khanh Nguyen Di

REVIEWER # 2

This paper contains important findings in terms of investigating medical professionals' perception of antimicrobial resistance, which is a global problem, and linking it to education. In this study, the number of participants is small scale, and its evaluation is limited, I believe that there is no problem in present form, and this paper could be accepted.

Thanks for your comment and acceptance.

Author Response

December 18th, 2022

To: Healthcare

Dear Editor and Editorial staffs,

We would like to send the revision letter regarding to the manuscript healthcare-2112814 entitled “Physician’s perspectives on factors influencing antibiotic resistance: a qualitative study in Vietnam”, submitted to Healthcare,by Khanh Nguyen Di et al.

We sincerely appreciate the hard work of the journal and the reviewers, which definitely make our manuscript become better. Please find the responses to reviewers’ comments in the next pages of this letter. All changes were made under “track change” mode in Word processing program.

Thank you very much for your consideration. In case more information is required, please feel free to contact us.

Best regards,

Khanh Nguyen Di

REVIEWER # 3

Introduction

The author should explain in the background section: the association of "Vietnamese can easily purchase antibiotics from the pharmacies", ''Only 27% of the pharmacy staff in Vietnam have correct knowledge about antibiotic use and resistance”, with the aim of this this study “the physician’s perspective on the factors influencing ABR in Vietnamese hospitals".

Thank you. We have revised this section, with more information and backgrounds, accordingly.

The author stated that: The findings of this study will help Vietnamese policy makers to plan and establish future interventions to improve antibiotic prescriptions in the healthcare facilities. - What is the healthcare system in Vietnam? Are physicians in hospital can affect antibiotic use by people in the community?

Thanks for your question. Yes, the physicians in hospital could affect and influence the way of using antibiotic of the people in the community. We have added these information in the revised manuscript accordingly.

Methods

The author should explain how to develop themes and subthemes of the questionnaire and its validity. See reference: https://www.cdc.gov/antibiotic-use/coreelements/index.html#:~:text=CDC's%20Core%20Elements%20of%20Antibiotic,patient%20safety%20and%20improve%20outcomes.

Thank you, the themes/subthemes, the interview guide, and the questionnaire validity was developed based on our preliminary studies and the literature. We also conducted a pilot test to confirm the validity of the questionnaire. These information have been added in the revised manuscript.

This is inconsistency: the background: To date, the physician’s perspective on the factors influencing ABR in Vietnamese hospitals has not been investigated; but the participants: Participants in each group were healthcare practitioners from both outpatient care and inpatient care.

Thank you. However, we are sincerely sorry in case we misunderstand your point. Maybe we overlook it, but we could not see any inconsistency in these two sentences. In the first sentence, we pointed out that the opinions of the physicians on the ABR in Vietnam have not been investigated. In the second sentence, we recruited the healthcare physicians from the hospitals to participate in our study.

Results and Discussion

The author is supposed to present and discuss the hospital (inpatient) context and community

(outpatient) separately to be able to provide recommendations in both settings.

Thanks for your suggestion. We sincerely apologize in case we misunderstand your point. However, in this study, we focused on the physician’s perspectives, which mainly related to the hospital context. For the community (outpatient) settings, we put limited attentions, with the section 3.2.1 is the main context. The study on the community perspectives has been conducted by our group and will be published in the future. Therefore, we humbly think that we should focus on the physician viewpoints in this study. Thanks very much for your understanding. 

Conclusion

The author is supposed to state which physician's perspective is relevant to finding the solution and provide insights into future education.

Thank you, we have adjusted the conclusion accordingly.
